# Identification and Characterization of Multiple Paneth Cell Types in the Mouse Small Intestine

**DOI:** 10.3390/cells13171435

**Published:** 2024-08-27

**Authors:** Steven Timmermans, Charlotte Wallaeys, Natalia Garcia-Gonzalez, Lotte Pollaris, Yvan Saeys, Claude Libert

**Affiliations:** 1VIB Center for Inflammation Research, 9052 Ghent, Belgium; stevent@irc.vib-ugent.be (S.T.); charlotte.wallaeys@hotmail.com (C.W.); natalia.garciagonzalez@irc.vib-ugent.be (N.G.-G.); lotte.pollaris@irc.vib-ugent.be (L.P.); yvan.saeys@ugent.be (Y.S.); 2Department of Biomedical Molecular Biology, Ghent University, 9052 Ghent, Belgium; 3Department of Applied Mathematics, Computer Science and Statistics, Ghent University, 9000 Ghent, Belgium

**Keywords:** Paneth cells, single-cell RNA-seq, small intestine

## Abstract

The small intestinal crypts harbor secretory Paneth cells (PCs) which express bactericidal peptides that are crucial for maintaining intestinal homeostasis. Considering the diverse environmental conditions throughout the course of the small intestine, multiple subtypes of PCs are expected to exist. We applied single-cell RNA-sequencing of PCs combined with deep bulk RNA-sequencing on PC populations of different small intestinal locations and discovered several expression-based PC clusters. Some of these are discrete and resemble tuft cell-like PCs, goblet cell (GC)-like PCs, PCs expressing stem cell markers, and atypical PCs. Other clusters are less discrete but appear to be derived from different locations along the intestinal tract and have environment-dictated functions such as food digestion and antimicrobial peptide production. A comprehensive spatial analysis using Resolve Bioscience was conducted, leading to the identification of different PC’s transcriptomic identities along the different compartments of the intestine, but not between PCs in the crypts themselves.

## 1. Introduction

The inner lining of the intestinal tract, from the duodenum to the rectum, is covered by intestinal epithelial cells (IECs), which form a physical barrier between the (sterile) host and the intestinal content, which is considered as part of the external environment. The intestinal epithelium undergoes continuous renewal to preserve the integrity of the gut, which is potentially undermined by daily challenges posed by the microbiome and dietary components, which may contain a big amount of different molecules, metabolites, antigens, toxins, etc. [1,2]. Intestinal stem cells (ISCs) are responsible for this renewal and the constant replenishment of the epithelium by directing the generation and differentiation of different specialized IECs. This is performed in such a way that (i) the cell–cell contact, which is essential in the preservation of the physical barrier, remains intact, and (ii) the relative amounts of different IECs are preserved. These cells include absorptive enterocytes as well as four secretory cell types: enteroendocrine cells, goblet cells (GCs), tuft cells (TCs) and Paneth cells (PCs). The latter are exclusively found in the small intestine, namely in the crypts of Lieberkühn, which are deep well-like structures found in between villi [3].

In the small intestine, the epithelium consists of a monolayer of columnar IECs, with an architecture characterized by upright villi interspersed by the crypts of Lieberkühn. From the rostral to the caudal, the small intestine is classically divided into the duodenum, jejunum, and ileum. Along these segments, gradients of distinct environmental conditions have been observed; for example, the amounts of bacteria [4] and their diversity and composition [5], digestive enzymes [6], metabolites [7], etc. Recent research studying regional variations along the gastrointestinal tract has identified the crucial role of location-specific transcripts [8,9].

Regional gradients may have a significant impact on the IECs, in terms of required functions and genes to be expressed. For example, the increased amounts of bacteria in the ileum compared to the jejunum/duodenum correlate with an increased concentration of antimicrobial peptides (AMPs) and pH [10,11,12,13]. Also, mucus thickness on top of the IECs varies and is highest in the ileum [14]. Oxygen levels decrease along this trajectory [15]. Also, the duodenum contains the largest activity of digestive enzymes [6] and concentration of primary bile acids [16], which originate from the pancreas and liver, respectively. The nutrients from the partially digested food are primarily absorbed in the jejunum, while the ileum is responsible for absorbing residual content and specific vitamins, as well as reabsorbing bile salts. The ileum is thought to play a more pronounced role in contributing to immunological processes such as probing and regulating the microbiota composition and training of the immune system, and therefore is particularly rich in Peyer’s patches [17].

While the effects of environmental gradients on digestion have been well characterized [18], our understanding of how these gradients specifically influence the functionalities of the IECs and the microbiome, and vice versa, remain incomplete. A recent single-cell RNA-seq survey on the different compartments of the small intestine provided interesting insights into cellular diversity and subpopulations of different cell types in mice. Notably, it was shown that PC gene expression is quite distinct from other IECs, and that there is evidence to suggest that there are two different PC subtypes, linked to the regional variations of these cells over the course of the small intestine [19].

PCs are positioned in the crypts of the entire small intestine and their quantity increases steadily from the proximal to the distal side, reaching an average of three times higher levels in the distal regions, correlating with increased bactericidal activity [20]. The primary function of PCs is secreting AMPs (mainly α-defensins and lysozyme) and providing support to the Lgr5^+^ ISCs [2,21]. Paneth cells can also contribute to the intestinal regenerative response to irradiation by de-differentiation and gaining stem cell-like characteristics [22]. Moreover, PCs have been shown to have additional functions, such as performing phagocytosis and efferocytosis, sequestering heavy metals, preserving barrier integrity, and driving inflammation [23,24,25,26,27]. Despite the multiple functions of PCs, these cells are often considered and studied as a homogeneous cell population. In this study, we identify multiple subpopulations (clusters) of PCs based on single-cell gene expression profiles on pure sorted PCs starting from the entire small intestine, some of which appear as tuft cell-like PCs, others as GC-like PCs, and still others as stem cell-like PCs. Using bulk RNA-seq on pure sorted PCs from the three major compartments of the small intestine, more in-depth knowledge is obtained of the regional variations in gene expression and functions and possible active master regulator transcription factors that contribute to the distinct PC transcription profiles and clusters. By combining the single-cell data, bulk RNA-seq data, as well as spatial data obtained by Resolve in situ hybridization technology, we were able to link some clusters with regions in the different compartments of the small intestine, and hence with specific functions via pathway analysis. We also investigated any possible link between expression patterns and places in the crypts. The data suggest that the complexity and diversity of the gut microenvironment are reflected in the existence of multiple PC types and they open many new lines for further research.

## 2. Materials and Methods

### 2.1. Mice

C57BL/6J mice were purchased from Janvier Labs (Le Genest, France). The mice were housed in individually ventilated cages, maintained under a 12 h light/dark cycle within a specific pathogen-free facility, and provided unrestricted access to food and water. Female mice of 11 weeks were used. All experiments were approved by the local Ethical Committee.

### 2.2. PC Sort

Mice were euthanized and single-cell suspensions of isolated small intestinal crypts were made as previously described [19]. Cells were labeled for 30 min in the dark on ice with the following fluorescently conjugated antibodies diluted in HBSS + 2% FCS: APC anti-mouse CD24 Antibody (1/125, 101814, BioLegend, San Diego, CA, USA), PE anti-mouse CD117 (C-Kit) Antibody (1/250, 105808, BioLegend), Brilliant Violet 421TM anti-mouse CD31 Antibody (1/100, 102424, BioLegend), Brilliant Violet 510TM anti-mouse CD45 Antibody (1/100, 103138, BioLegend), Brilliant Violet 421TM anti-mouse TER-119/Erythroid Cells Antibody (1/100, 116234, BioLegend). Flow cytometry was conducted using a BD FACSAria™ III Cell Sorter equipped with a 100 μm nozzle and BD FACSDiva™ software (8.0.1, BD biosciences, Franklin Lakes, NJ, USA). Fluorophore voltages were optimized before each experiment, with daily CST verification. To eliminate debris and doublets, a sequential gating approach was applied. Initially, gating based on forward scatter area versus side scatter area was used, after which gating was refined by evaluating forward scatter width versus forward scatter height, and finally, by considering side scatter height versus side scatter width area. Viable cells were determined by excluding 4,6-diamidino-2-phenylindole (Molecular Probes)-positive cells. PCs were sorted as CD24^+^C-kit^+^SSC^Medium-High^ CD31^−^CD45^−^TER119^−^ cells and collected in 1 mL sort buffer (50% advanced DMEM/F-12 and 50% FCS). Afterwards, cells were washed with PBS and resuspended in 350 μL RLT buffer containing 1% β-mercapto-ethanol and stored at −80 °C.

### 2.3. RNA Isolation on Bulk PCs

RNA isolation on PCs (15,000 cells per sample) was performed using the RNeasy Plus Micro Kit (Qiagen, Hilden, Germany, 74034) according to manufacturing instructions. Sample homogenization before the start was achieved through 1 min of vortexing.

### 2.4. PC Bulk RNA-Seq

The RNA was used for creating an Illumina sequencing library using the Illumina TruSeqLT stranded RNA-seq library protocol (VIB Nucleomics Core, Leuven, Belgium) and single-end sequencing to 100 bp was completed on the NovaSeq 6000. The obtained reads were mapped to the mouse reference transcriptome/genome (mm39/gencode v28) with STAR (2.7.10b), and read counts were obtained during alignment using the STAR “--quantMode GeneCounts” option. Differential gene expression was assessed with the DESeq2 package (v 1.42.1) [28], with the FDR set at 5%. Differentially expressed genes were clustered with k-means clustering in R based on their normalized and scaled expression patterns. We tested values 2 to 15 for the number of clusters and used the gap measure, elbow plots, and silhouette analysis to determine to finally proceed with k = 7 for further analyses. The genes in each group were then used in functional enrichment analyses using Metascape (v 3.5.20230417) [29]. All RNA-seq data can be found in the public domain with accession number GSE255507.

### 2.5. PC scRNA-Seq

PCs were obtained by use of the PC sort protocol and immediately processed for single-cell droplet sequencing (10X genomics) by the VIB single-cell core according to the manufacturer’s protocol. We obtained reads as demultiplexed fastq files which we processed by the Cell Ranger pipeline (10X genomics cell ranger 7.1.0) [30]. To allow for comparisons with our bulk data, a custom cell ranger index was created from the mm39 reference genome and gencode structural annotation v28 release according to the “build a custom reference” guide by 10X genomics. Cell Ranger outputs, specifically the *filtered_feature_bc_matrix*, were further analyzed in Seurat v4.3. First, we filtered the dataset using min cells 10, min features 100, max features 2500. Further selection was completed based on the proportion of mitochondrial gene expression per cell. A maximum proportion of mitochondrial genes of 5% was taken as cut-off for retention of cells. Data were log-normalized and scaled to 10,000 cells, after which a search for variable features was completed (n_features = 2000). To minimize effects from cell cycle as a confounding factor but not remove real variance as our data may contain proliferating cells, the effects of the cell cycle phase difference were regressed out of the dataset using the cell cycle dataset [31] included in Seurat. Based on jackstraw analyses, we selected the first 15 principal components to find neighbors and perform clustering. Markers genes were obtained by the ‘FindAllMarkers’ function, with a minimal proportion of 60% detection in at least one cluster and a 10% significant difference. Using the significant positive markers per cluster, a functional analysis was performed using the g:Profiler2 (0.2.3) R package [32]. All visualization was performed with either Seurat v4.3 (functions vlnplot, featureplot, dotplot, and dimplot) or directly with ggplot2 (3.5.1). GEO accession numbers of the sequencing results: GSE273983.

### 2.6. Bulk and scRNA-Seq Comparison

Comparison and integration of bulk RNA-seq and scRNA-seq data was performed in R. Considering the data, each region from the bulk RNA-seq comprises a mixture of cells from clusters from the scRNA-seq. We constructed pseudo bulk data from the scRNA-seq at the cluster level by aggregating the expression of the individual cells. Fully aggregated sets were constructed by also aggregating the clusters onto one set. We used correlation analysis between the bulk region data and the scRNA *pseudo bulk* sets as well as generalized linear models (GLMs) to find links between both datasets. K-means clustering: to determine the optimal number of clusters, we used silhouette width analysis and the gap measure statistic. Metrics were obtained for a range of k from 1 to 15 using the factoextra library function ‘fviz_nbclust’. The global optimal number of clusters is equal to the number of sample intestinal regions, but to obtain a finer granularity, we selected the second optimum in the range at k = 7

### 2.7. Visualizations

All data visualizations, except those related to *Metascape* analyses, were created with the R ggplot2 (3.5.1) [33] package, unless otherwise stated, and arranged into multi-panel figures with the ggpubr (0.6.0) [33] and cowplot (1.1.3) packages. For the bulk RNA-seq, the heatmap was created with clustviz (0.0.0.9) [34]. All functional enrichment graphs were generated by Metascape.

### 2.8. Tissue Preparation Molecular Cartography

Tissue preparation was performed according to manufacturing instructions from Resolve (Tissue preparation protocol v1.5). In short, the duodenum, jejunum, and ileum were isolated, flushed with PBS, and placed directly in pre-cooled isopentane of −50 °C for 45 s. Afterwards, the tissue was imbedded in Tissue-Tek^®^ O.C.T. compound and stored at −80 °C. Cryoblocks were sectioned at 10 μm using the CryoStar NX7 and carefully placed on a molecular observation slide. The duodenum, jejunum, and ileum of 4 different C57BL/6J mice were sampled.

### 2.9. Molecular Cartography Probe Design

The probe panel was designed based on data from the single-cell study and bulk RNA-seq on purified PCs from the intestinal regions (this paper) along with the cell type expression data provided by Resolve biosciences and in-house bulk RNA-seq from PCs (GSE269510, GSE237588, GSE237759, GSE267927, GSE267790). We selected probes for all cell types we might find in the intestine, with a primary focus on PCs, including differential markers for the clusters found in the scRNA-seq, GCs, and stem cells. Taking into account the technological constraints, an initial set of potential probes was filtered from the data programmatically where cell type-specific genes were selected and filtered on minimal transcript length, and minimal and maximal expected signal. These limitations had a significant impact on the choice of genes for the probe set. Genes that were too short or had insufficient unique bases could not be used. According to Resolve’s standards, α-defensins, which would have been most interesting, were unusable for the probe panel. Furthermore, genes with very high expression were also unsuitable for probe design, which excluded genes such as *Lyz1*. This set was then checked with the Resolve biosciences data to further filter out unsuitable probes and to prevent the total expected signal from all probes combined from exceeding the maximum detection limit. This resulted in a final set of 59 probes (see Appendix A).

### 2.10. Molecular Cartography^TM^

Resolve Molecular cartography was performed on fresh frozen sections. The molecular cartography images and data were investigated with ImageJ (2.1.0) and the molecular cartography plugin for quality and simple visualizations and comparisons. For the cell-based analysis, we used an in-house pipeline. Data processing for cell-based analysis comprised two main steps: cell segmentation and transcriptomic analysis. To allow cell segmentation with the cellpose algorithm, the images were first pre-processed to improve the signal-to-noise ratio. This consisted of joining together the discrete tiles and processing the image with skicit-image. Processed DAPI-stained images were segmented with cellpose with an expected nuclei diameter of 30 pixels. The effectivity of the segmentation varied between the images, but we were able to identify and segment the majority of nuclei in each image. Nuclei were enlarged with the Voronoi method to include more data. Transcripts were then assigned to the segmented nuclei, allowing for single-cell processing of the data while maintaining the spatial information. The Python (3.10.8) package Scanpy (1.9.4) was used to process the data further as a single-cell dataset. We performed all clustering and analyses on the first 17 principal components and used a spread of 1.0.

### 2.11. Defa21 ELISA and Lysozyme Activity Assay

Ileum samples were lysed in PBS using the Qiagen TissueLyser II (5 min; 20 Hz). Samples were centrifuged at 10,000× *g* for 5 min and a Defa21 ELISA and Lyz1 Activity were tested on the supernatants. To measure protein concentration of Defa21 (abx510182, abbexa, Cambridge, UK), the protocol was performed according to manufacturing instructions. Lysozyme activity within the ileum was measured via the EnzChek™ Lysozyme Assay Kit (E22013, Thermofischer, Waltham, MA, USA), according to manufacturing instructions.

## 3. Results

### 3.1. Single-Cell Analysis Unveils Diverse PC Subpopulations

A total of 20,343 PCs were purified from the entire small intestine of 3 C57BL/6J mice using the FACS purification protocol as specified (setup in Appendix A) and based on CD24 and C-kit double-positive staining, as described [35]. Low-quality cells were filtered out during processing with the Cell Ranger–Seurat pipeline [36], yielding a total of 15,152 cells for subsequent analyses. This provided unique insights into potentially different PC populations in vivo in mice. During the analysis, we first controlled the expression of known PC markers in the cells. Lysozyme 1 (*Lyz1*) [37], a well-established PC-specific marker, exhibited expression in nearly all cells (except in 12 cells, i.e., 0.06%) as shown in Figure 1A. Likewise, we examined the expression of the PC marker *Clps* [38] (Appendix A), FACS sort marker genes *Cd24a* and *Kit* (Appendix A), and members of the defensin-α (*Defa*) [39] gene family (Figure 1H,I,K).

We performed unsupervised graph clustering using the *Leiden* algorithm, revealing a total of nine clusters, which themselves fall into three distinct superclusters on the UMAP projection (Figure 1B). Notably, *Lyz1* was expressed across all clusters. Although the expression levels were not uniform, every cell was positive for *Lyz1*. Clusters 5 and 8 exhibited the lowest expression, whereas the other clusters showed very high expression levels (Appendix A).

Discriminating markers per cluster (that help to define the clusters) were identified, and Figure 1C illustrates the top five markers for each cluster. Marker genes refer to genes that exhibit significant differential expression in a specific cluster when compared to the mean expression across all other clusters. The amount of differentially expressed (DE) markers per cluster is displayed in Figure 1D and a list of upregulated and downregulated marker genes per cluster can be found in the addendum as Appendix A. The number of genes detected on average in a cell per cluster varies and is a contributing factor to the clustering. Given that the gating strategy (see Materials and Methods) maximizes doublet exclusion and that we performed a check for doublets, there is only a low chance that this is an artifact and not biologically meaningful.

We characterized all nine clusters based on marker gene expression and found several clusters expressing markers not identified as PC markers. The tuft-2 cell markers *Dclk1* [40] (Figure 1E), *St18* [41], and *Rgs13* [41] can be found highly expressed in cluster 8. Tuft cells are IECs that play a critical role in immunity against parasite infection, or, in case of tuft-2 cells, bacterial infections. Likewise, the GC marker *Fcgbp* [42] is expressed by the majority of cells in cluster 5 (Figure 1F), which also highly expresses other GC markers such as *Tff3* [43] (Appendix A), *Agr2* [44,45], and *Lgals2* [31]. Considering that the cells in these two clusters also express PC markers, we do not consider them as contaminants, but rather as specific PC subpopulations sharing characteristics with other intestinal cell types. Clusters 8 and 5 were thus assigned as tuft cell-like PC (TC-PC) and GC-like PC (GC-PC), respectively.

Cluster 6 displays high expression of Olfm4 (Figure 1G), a marker associated with immature cells [46]. Moreover, 35% of the cells in this cluster show moderate expression of the stem cell marker *Lgr5* [47] (Appendix A). Therefore, we believe that cluster 6 comprises immature PC progenitors that are on the early stage of differentiating from stem cells and have not yet lost all stemness transcript markers.

We also compared our data with the dataset of Haber et al., specifically focusing on their identified marker genes. This was performed to determine contamination based on established data and to focus on our cell populations that co-express markers from non-PC types together with PC markers. Haber et al. identified cell type-specific markers for a wide variety of cells from the small intestine. For our cells with tuft-like characteristics, GC characteristics, and immature progenitor PCs, we can confirm the presence of several markers for tuft cells, GCs, and stem cells from Haber et al., with the latter being the least prevalent. To go into detail, we found very low to no detection of most mature proximal enterocyte markers such as *Apoa4* (Figure 2A) and *Fabp1* (Figure 2B); some markers can, however, be found in almost all clusters such as *Krt8* or *Apoc2* (Figure 2C). Likewise, we could not find appreciable expression of distal enterocyte markers in our dataset, such as *Tmigd1*, *Fabp6*, or *Slc51b* (Figure 2D–F). In contrast, several goblet cell markers are found in all clusters, including *Agr2* (Figure 2G) and *Klk1* (Figure 2H), but they are more highly expressed in cluster 5. Some GC markers described by Haber et al. are present in very few cells (e.g., *Zg16*, Figure 2I). This lends support to the absence of typical GCs in our purified cell population. The tuft cell markers can be found, but only cluster 8 shows consistent detection and higher expression levels (Figure 2J–L). Additionally, our purified cell population contains nearly no genes of enteroendocrine origin. We can confirm the presence of typical Paneth cell markers in all our clusters, including those identified by Haber et al. [19].

Overall, we can find two types of discriminating markers (Appendix A). The first are those that are universally expressed, such as *Lyz1* and the *Defa* family, which help define all total purified cell populations and provide some subpopulation discriminatory power, based on the expression level per cluster. A combination of different marker expression profiles is needed to identify a cluster. The second are several more specific genes that are expressed in only one cluster or limited to a few clusters, and have more discriminatory power (e.g., *Dclk1*).

Among the IECs, PCs are unique in the expression of “*Defa* gene family members”, a set of highly conserved 2-exon genes that encode α-defensins with antimicrobial functions [48,49]. The genes, numbered from *Defa1* to *Defa43*, are located on chromosome 8 [50]. However, the numbering is not continuous, and 39 *Defa* genes are annotated in the MGI database. They are expressed in all sequenced cells. A combination of the *Defa* gene expressions serves as the primarily discriminating markers for clusters 0 to 4 compared to the other clusters, which indicates that these are mature PCs with classical AMP functionality, as described in previous studies [51]. They also allow some discrimination between these clusters themselves, as for example, *Defa21* is expressed most highly in cluster 0, only slightly lower in 2 and 4, and lower in cluster 1 and 3 (Figure 1H). While *Defa39* is expressed the highest in cluster 1, the differences are small (Figure 1I). This indicates a difference in the antimicrobial activity among the PC subpopulations and might correspond to some degree to the proximal–distal intestinal axis. Interestingly, despite filtering out cells showing a high proportion of mitochondrial gene expression, we detected mitochondrial-encoded genes as cluster markers, including *mt-Cytb* (Figure 1J), in clusters 5, 6, and 8, and especially in cluster 2, for which they are highly discriminating markers. Cluster 4 is one of the PC clusters that shows its own signature of α-defensin, with the *Defa34* gene strongly expressed in many cells (Figure 1K). Finally, cluster 7 does not exhibit any specific positive markers. However, it is characterized by several negative markers, particularly the mitochondrial genes that are highly expressed in cluster 2 (Figure 1J,L).

The top 100 positive marker genes of each cluster were subjected to functional analysis using Metascape (v 3.5.20230417) [29] and g:Profiler2 (v 0.2.3) [32]. Partially due to the overlap of markers between clusters, establishing a unique functional profile for each cluster was not possible. However, the analysis identified several distinctive functional groups. The results from the functional enrichment provide additional insight into the marker-based cluster identification. The clusters 0 (Appendix A), 1 (Appendix A), 3 (Appendix A), 4 (Appendix A), and 7 (Appendix A) showed strong enrichment in pathways related to mucosal immunity and the defense response against bacteria and their products, indicating functions of mature PCs. These can be further subdivided in clusters with a strong prediction for direct defense responses to bacteria (clusters 0 and 7) and clusters which also show more general immune response pathways (1, 3, and 4). Cluster 2 (Appendix A) displayed enrichment for mitochondrial functions, mitochondrial respiration, and ATP production. Clusters 5 (Appendix A), 6 (Appendix A), and 8 (Appendix A) were marked by pathways enriched for protein translation activity, which may indicate an expansion of protein production capacity in preparation for a secretory function. This functional enrichment was primarily driven by the expression of ribosomal proteins.

In conclusion, the markers and functional enrichments indicate that clusters 0, 1, 3, and 4 are mature PCs, expressing the largest amount of α-defensins. We found a group of immature PC progenitors which are grouped together as cluster 6 and characterized by high *Olfm4* and some *Lgr5* expression. We also discovered a subset of cells that have tuft cell characteristics found in cluster 8 (TC-PCs) and PCs with GC properties (GC-PCs), primarily found in cluster 5. Two clusters were hard to further classify, with cluster 2 expressing several mitochondrial genes and cluster 7 defined by the very low expression of mitochondrial chromosome genes and moderate–high level of α-defensins. It is possible that these clusters mark senescent or even dying cells, as these are often enriched for mt-RNA products in single-cell sequencing, but we cannot exclude a biological role.

### 3.2. Regional Variations in PC Transcriptomes along the Small Intestine

To gain deeper insights into location-dependent transcriptome variations in mouse PCs and how these may correlate to the clusters that were detected in the single-cell analysis, we performed a bulk deep RNA-seq on PCs isolated from three small intestinal regions. In contrast to the full intestine single-cell strategy, we opted for a bulk strategy as data from many mice needed to be pooled, and we aimed for an in-depth view of the transcriptome, which is impossible with single-cell sequencing. An overview of the setup and analysis can be found in Figure 3A. In brief, we took a small part of the duodenum, jejunum, and ileum of C57BL/6J mice and sampled and pooled PCs of eight mice in order to obtain sufficient cells per sample (15,000). Standard pipelines for the identification of differential gene expression (see Materials and Methods) were used to reveal a PC transcriptome that depends on the location along the small intestinal tract. Genes were grouped according to expression profile and groups were analyzed for functional enrichment.

When comparing the three regions pairwise, PCs from the duodenum and ileum showed the largest differences, with 1549 differentially expressed genes (DEGs) between them; those of the duodenum and jejunum showed only 283 DEGs, while the number of DEGs in PCs from the jejunum and ileum was 600. Overall, we found that a total of 1674 genes are differentially expressed (union overlap), so the PCs from the three regions have sufficient distinct gene expression profiles that allow them to be easily discriminated from one another (Appendix A).

We used K-means clustering to group the gene expression profiles of the DEGs. As this requires that the number of clusters (K) is set a priori, we tested several values for K and evaluated the clustering using various metrics: *average silhouette size* (Appendix A) and *the gap statistic* (Appendix A, see Materials and Methods). Based on the data, we selected a value of 7 for the number of desired clusters. These seven groups fall into three main categories: three groups of genes that are most highly expressed in the duodenum (groups 1–3, Figure 3B–D), one group of genes most highly expressed in the jejunum (group 4, Figure 3E), and three groups of genes expressed most highly in the ileum (groups 5–7, Figure 3F–H). Each of these groups were subjected to functional *Metascape* enrichment analysis and the top five pathways for each group are displayed in Figure 3I (individual *Metascape* analyses can be found in Appendix A and the full *Metascape* analysis is in Appendix A). The full gene lists corresponding to each group are added to the addendum, Appendix A.

Genes highly expressed in duodenal PCs consist of three groups.

Group 1: the group of genes displaying the highest expression in the duodenum and nearly no expression in the ileum displayed an expression profile reminiscent of enteroendocrine cells and genes associated with pancreas development (Figure 3B), suggesting that PCs in the duodenum may play a role in the regulation of hormonal signaling and in coordinating digestive processes, respectively. Pancreatic and duodenal homeobox 1 (Pdx1) is a gene belonging to this first group. Although its function in PCs is not known, it plays a key role in the maturation and preservation of pancreatic β-cells, as it controls the activation of insulin and other genes responsible for glucose sensing [52]. Group 2: genes having a slightly lower duodenal expression and a less severe drop-off in expression level along the intestinal axis (Figure 3C) are related to the secretion of corticotropin and the catabolism of both sterol and cholesterol processes. This suggests that PCs, particularly those in the duodenum, may play a role in both hormone secretions related to stress responses and the degradation of specific lipid molecules. Group 3: these genes (Figure 3D), which show expression in all compartments with a proximal–distal decreasing pattern, are related primarily to the cell cycle. This suggests that cell proliferation also follows a proximal–distal axis with the duodenum showing slightly more cell cycle activity.

Genes most highly expressed in jejunal PCs are enriched for amyloid-beta clearance, fat absorption, ER stress, and cholesterol metabolism (group 4, Figure 3E). The precise role of secreted amyloid-beta in the gut is not fully known; however, it has been observed to attenuate cholesterol uptake in intestinal epithelial cells in vitro [53].

Genes most highly expressed in ileal PCs also consist of three groups. Group 5: the genes ubiquitously expressed in PCs, but increasingly along the GI tract and most highly in ileum PCs, are primarily associated with secretory pathways (glycosylation processes, Golgi complex) and metabolism (Figure 3F), compatible with secretory cells. Group 6: these genes (Figure 3G) display a clear enrichment for antimicrobial and immunity-related functions and the expression of these genes is very high in ileum but falls quickly towards the proximal part of the SI. This correlates with the observations that immune reactions [17] and antimicrobial defense [20] are higher in the ileum, likely correlating to the increasing level of bacteria along the small intestine [13]. We indeed observed the highest concentration of Defa21 in the ileum (and jejunum) of mice, and significantly lower levels in the duodenum (Appendix A). No differences were observed in lysozyme activity between the three compartments, which is in line with the equal expression of lysozyme in the three compartments (Appendix A). The last group, group 7, which is highly specific for ileal PCs and has lower functional enrichment, was associated with pathways such as chloride transport and the positive regulation of cholesterol metabolic processes (Figure 3H).

Beyond pathway enrichment, Metascape also provided a set of predicted upstream regulators for four out of the seven groups, namely the largest groups, revealing a highly distinct regulatory profile (Appendix A). Group 2 DEGs are regulated by specificity protein 1 (*Sp1*) and nuclear receptor coactivator 1 (*Ncoa1*). Group 3 genes are regulated by transformation-related protein 53 (*Trp53*), transformation-related protein 57 (*Trp57*), and Achaete-scute homolog 1 (*Ascl1*). Group 5 genes, ileum-enriched, appear to be regulated by Fos proto-oncogene (*Fos*), CCAAT enhancer binding protein beta (*Cebpb*), CREB-binding protein (*Crebbp*), androgen receptor (*Ar*), and growth factor-independent 1B transcriptional repressor (*Gfi1b*). Group 6, slightly enriched in the ileum, was regulated by regulatory factor X5 (*Rfx5*), regulatory factor X-associated ankyrin containing protein *(Rfxank)*, and regulatory factor X-associated protein (*Rfxap*), which are transcription factors involved in the activation of the major histocompatibility complex (MHC) class II. This is in line with the elevated immune response (mediated by AMPs) pathway, observed within this group.

Overall, on the transcriptional level, we observe that AMP expressions in PCs are lowest in the duodenum and increase along the GI tract to reach their maximum in the ileum. In contrast, functions related to digestion and non-AMP secretions seem to be present primarily in PCs in the duodenum and jejunum, and less in the ileum. Although these are all the same cell type, location-dependent variations clearly exist, and these correlate with the proximal–distal axis and the gradients, such as luminal nutritional content, luminal pH, and luminal bacterial load. These differences may be governed at the cellular level by the activation of distinct master regulator transcription factors, which are possibly instructed by these environmental gradients and other factors, and lead to location-specific transcription profiles and fine-tuned cell functions.

### 3.3. Integrative Analysis of Bulk and Single-Cell Data Links Specific PC Clusters to Regions in the Small Intestine

To integrate bulk and scRNA-seq data, the use of classical bulk deconvolution methods (e.g., “*bisque*”) is not feasible or recommended and is discarded for the following primary reason: reliable methods use a public database related to cancer and immune cell types or required multiple scRNA-seq samples as a reference [54]. Our single sample (though retrieved from eight mice) is insufficient to create a PC database. Based on how the classical deconvolution methods work, we decided to perform direct comparisons of the bulk and scRNA-seq data based on the intersect of detected differentially expressed genes using “*correlation analysis*” (Figure 4A) and “general linear model” (GLM) regression tools. Through these tools, we observed clear links between several scRNA clusters, bulk sampling regions, and groups. Most clusters exhibited high to very high correlations within all regions, typically exceeding a correlation coefficient of 0.7. In general, all regions are found to contain signals, and thus cells from each scRNA cluster. However, some clusters are more specifically linked.

scRNA clusters 0, 4, and 7 had a very high correlation score with the ileum but a considerably lower score, especially for cluster 0, with the other regions, the duodenum in particular. In contrast, cluster 1 and cluster 3 exhibited very high correlation scores with both the jejunum and duodenum; however, their correlation with the ileum was weaker. Cluster 2 showed a very strong correlation with the jejunum and a good correlation with the duodenum and ileum. This suggests that clusters 0, 4, and 7 consist of ileal PCs, and clusters 1, 2, and 3 represent the more proximal PCs (Figure 4B). The remaining scRNA clusters displayed relatively uniform correlations with the different intestinal regions. The highest correlations were observed in the jejunum, followed by slightly reduced correlations in the duodenum and lower correlations in the ileum. These findings indicated that some clusters, found in the single-cell dataset, are influenced by the proximal–distal axis, meaning that some clusters are derived more from a single region of the intestine, while others are not influenced and comprise cells that are evenly distributed along the proximal–distal axis.

The expression pattern of a selection of individual marker genes, either from the literature or from the scRNA-seq data in the bulk sequenced regions, allowed us to zoom in deeper (Figure 4C).

scRNA cluster 8, which was considered tuft cell-like PCs, demonstrated similar correlation scores across the three regions: we observed that the typical markers were present at a low level in the bulk RNA-seq for all regions (average expressions: *Dclk1* = 30.36, *St18* = 1.26, *Rgs13* = 17.74) and were not differentially expressed between any of the intestinal regions (Figure 4D). Consequently, cells from cluster 8 appeared to be evenly distributed along the small intestine and all regions contributed equally to the signal.

Cluster 5, considered GC-like PCs, was more challenging to correlate, because either not all cells express the marker (60% of the cells are *Fcgbp*-positive) or the marker is ubiquitously expressed, but at a low level (*Tff3*, Figure 4E). Interestingly, the most ubiquitously expressed marker, *Tff3*, displayed no difference in the bulk RNA-seq data, while *Fcgbp* was significantly more highly expressed in the duodenum. This could indicate that there are multiple populations of GCs, that only some of them express high *Fcgbp*, and that these populations decrease along the intestine or that the expression of *Fcgbp* in this cell type overall decreases towards the distal intestine.

Cluster 6 is defined by the *Olfm4* marker, which is a characteristic of immature intestinal cells [55]. The contribution of this group of cells to the intestinal regions appeared to be region-dependent. The expression of *Olfm4* was approximately two-fold lower in the ileum compared to the duodenum, although the difference did not reach statistical significance (*p* = 0.08, Figure 4F). This could mean that either the total number of Olfm4^+^ cells is lower in the ileum compared to the total number of PCs/crypt cells, or that they express the marker more lowly than in the other regions, or a combination of both, which will be further investigated via spatial transcriptomics with Resolve molecular cartography.

Markers from the other clusters, such as *mt-Cytb* (Figure 4G), show very little difference in expression between the different intestinal regions, indicating that these cell populations are present to the same extent along the intestine and not affected by any proximal–distal axis gradient(s).

In conclusion (see also Appendix A), the majority of the clusters were assigned an identity. Purely based on scRNA markers, we could previously identify several clusters, such as the immature PC progenitors, GC-like PCs (two clusters), and tuft cell-like PCs. This can now be extended with information from the intestinal regions to also assign the identity of ileal PCs to clusters 0 and 7. Clusters 1, 2, and 3 comprised the more proximal PCs. To link the clusters to more specific regions, we combined our general correlation data and marker expression data with the known literature. It is known that several, but not all, Defa family members (including *Defa21*) have an increasing expression pattern along the small intestine, with ileal PCs displaying the highest levels of expression, which is confirmed in our dataset (Figure 4H). In contrast, duodenal PCs have lower expression of these genes, but express the *Rnase1* gene to very high levels. We also examined overall α-defensin expression in more detail. Among the 16 overall most highly expressed α-defensin genes, we identified five genes that share the same expression pattern as *Defa21* (including *Defa21* itself) and two additional genes, *Defa29* and *Defa5*, that exhibit a closely related pattern (with lower expression in the duodenum). Several genes, such as *Defa40* and *Defa3*, show a peak in expression in the jejunum. Additionally, we observed that some genes, e.g., *Defa39* and *Defa38*, have expression patterns nearly opposite to that of *Defa21* (Appendix A).

Based on these data and the correlation and regression results, we conclude that the duodenal PCs can be found in clusters 1 and 3 and the jejunum PCs in cluster 2. Cluster 4 seems to be made up of distal PCs, but we cannot say if they are from the jejunum or ileum, so it is classified as jejunum/ileum, see Figure 4B.

### 3.4. Confirmation of PC Clusters via Spatial Transcriptomics

#### 3.4.1. Spatial Transcriptomic Probe Design to Validate PC Clusters

The last approach to study PCs in depth consisted of performing spatial transcriptomics on the three different compartments of the small intestine (see Figure 5A for set up) using the Resolve molecular cartography platform of Resolve Biosciences. This tool allows the examination of the expression of several dozen RNA transcripts using a hybridization approach with a custom-designed probe panel in a manner that also detects subcellular spatial information. The goal was to confirm the existence of distinct PC populations along the gastrointestinal tract and to investigate if there are different PC populations within a specific crypt region. The technology allows for the imaging of up to 100 probes, but limitations of the maximal allowed total (predicted) signal intensity forced us to use fewer probes. Technical limitation also excluded *short* RNA molecules, including all *Defa* genes, and extremely highly expressed genes (e.g., *Lyz1*), leading to a final probe set of 59 probes (see Materials and Methods). Resolve utilizes DAPI staining for nuclei as the standard method to localize cells. Optionally, IHC for proteins can be included, which allows for the staining of a membrane marker. However, this is not guaranteed to be successful, and in our case, the included IHC markers failed to yield results suitable for further processing. Data were analyzed using simple proximity methods as a sanity check and processed with an in-house pipeline, which allows cell segmentation and cell-based analysis (Figure 5A).

#### 3.4.2. Simple Co-Location Analysis Confirms the Existence of Tuft Cell-like PCs and GC-like PCs

The initial analyses consisted of a location-based but cell-free analysis, i.e., comparing marker expressions across samples and marker co-occurrences. These were primarily used as quality controls for the obtained dataset and to validate the existence of tuft cell-like PCs and GC-like PCs. For example, *Rnase1* expression in the Resolve samples did indeed confirm the pattern of decreasing expression along the proximal–distal axis as found in the bulk RNA-seq data, and previously described in the literature [19] (Figure 5B). We also found a strong co-expression of *Rnase1* and PC markers *Hapb2* and *Nupr1*. The second observation that was made from the crypts is that the abundance of *Olfm4* (Figure 5C) also decreases along the intestine: in the duodenum, nearly all PC marker occurrences co-localize with *Olfm4* (Figure 5D), and this is also observed in the jejunum (Figure 5E). In contrast, this is no longer observed in the ileum due to the much lower expression of *Olfm4* (Figure 5F). Finally, we confirm that the GC marker *Fcgbp* and the tuft marker *Dclk1* can indeed be found in the crypts, next to their usual presence in the villi, and primarily in the duodenum. The occurrence of both of these subsets is very rare, but the Resolve data show several clear co-occurrences of such cells expressing either *Fcgbp* or *Dclk1* in intestinal crypts, together with PC markers (Figure 5D–F). This confirms the existence of tuft-like PCs and goblet-like PCs within the intestinal crypts, albeit in a limited quantity. The use of the DAPI staining helped in preventing the detection of false interactions but could not eliminate it, which is why we switched to a cell–nucleus segmentation approach for a more detailed analysis.

#### 3.4.3. Cell-Based Analysis Identifies Region-Dependent Paneth Cell Identities along the Small Intestine, but Not within the Crypts Themselves

In-depth cell-based analyses were performed using an in-house processing pipeline. This pipeline allowed for the following: (i) image pre-processing (see Materials and Methods), after which (ii) cell segmentation with the Cellpose algorithm could be performed. The identification of cells and (iii) assignments of the transcripts to those cells allowed us to (iv) use the data as a single-cell dataset and use the Scanpy tool for single-cell analyses while maintaining the link to the spatial information. Here, we will discuss the analysis of the data using the duodenum region of interest (ROI) 1 for representative visualizations. The images from all other regions can be found in the Appendix A repository Figshare (10.6084/m9.figshare.25999150).

Raw DAPI data (Appendix A) were evaluated and pre-processed (see Materials and Methods). The resulting cleaned image was used for nuclei segmentation using Cellpose (2.2.3) (example in Figure 5G). The detected transcripts were then assigned to individual segmented nuclei. However, since this step only retains transcripts that overlap with nuclear locations, only 20% (on average) of transcript information was retained, which proved insufficient for downstream analysis. To include more transcript information, the segmented nuclei were enlarged by 30 px, resulting in a retention of 40–50% of transcript information, up from 20%, and allowing further downstream processing. The data for each sample were then processed as a single-cell experiment, analogous to the scRNA-seq on purified PCs, but now using Python (3.10.8) and Scanpy (1.9.4) instead of R (4.3.1) and Seurat (4.3).

The results for each sequenced region of interest show a dense UMAP without any clearly separate groups and (depending on the size of the region) between 10 and 16 *Leiden* clusters, using the same parameters for each sample. These clusters can be annotated to intestinal cell populations with high or low granularity. For the first duodenum ROI, there were 12 clusters (Figure 5H) and these cluster identities could be mapped back on the DAPI image (Appendix A). We then saw cells expressing specific markers localized in a manner which agreed with our cell identification, such as the *Fcgbp* expressing GCs and the crypt-exclusive proliferating cells.

To enhance the discriminatory power for detecting various cell types in the crypts, all replicates per condition were merged. We verified the absence of batch effects in any of the sample combinations (duodenum: Figure 5I, jejunum: Appendix A, ileum: Appendix A). This resulted in a significantly greater number of detected clusters, also in the crypts, which now revealed at least two populations (in all intestinal regions). Unfortunately, the stem cells were not detected as an individual group in any region and were part of a group of mitotically active cells, i.e., grouped with the transit-amplifying cells. The tuft cells were also not consistently detected as a unique cluster; upon checking marker gene expression, they were spread out over multiple cell clusters. The identification of the PCs could be confirmed by mapping back the PC cluster on the original images (Figure 5J). We found between tow and six PCs per crypt, with the number of PCs increasing in the distal part of the intestine, as was described previously [56].

The ultimate resolution of this research would lead to annotating PC populations or individual cells in crypts based on scRNA expression. The identification of PC populations in Resolve *Leiden* clusters is a required first step, which was performed manually based on expressed markers (in the context of the probe panel). We focused specifically on cell types that were found in the scRNA and their markers: PCs, TC-PCs, immature PC/stem cells, and GC-PCs. As suggested by the scRNA-seq data, the following markers were chosen as primarily relevant in the Resolve dataset: *Nupr1* (PCs), *Habp2* (PCs), *Wnt3* (primarily PCs), *Rnase1* (proximal PCs), *Olfm4* (stem cell, immature PC progenitors), *Fcgbp* (GC-PC), and *Dclk1* (TC-PC). In the duodenum (Figure 6A), we identified, based on marker gene expression (Figure 6D), one cluster of PCs (9), one cluster of *Olfm4*-positive cells (0), a cluster of TCs (14), and two clusters of GCs (1 and 18). In the jejunum (Figure 6B), we identified one cluster of PCs (8), one cluster of *Olfm4*-positive cells (0), and three clusters of GCs (2, 13, and 20) from the marker gene expression (Figure 6E), and in the ileum (Figure 6C), we found, based on our marker gene expression in the *Leiden* clusters (Figure 6F), one cluster of PCs (4), one cluster of *Olfm4*-positive cells (weakly positive) (0), and three clusters of GCs (1, 18, and 21). The differences that can be found between individual PCs in the Resolve dataset are small and limited to proximal–distal axis effects. We were unable to find multiple PC subpopulations within the same area of the small intestine or detect any in-crypt variation of PCs. The PC populations in the duodenum, jejunum, and ileum are different in expression levels of PC-specific markers, such as *Nupr1*, which increase along the proximal–distal axis, and *Rnase1*, which shows the opposite. However, based on Resolve, we also found PC-specific probes that detected region-specific signals that were not previously detected in the single-cell or purified bulk datasets, namely *Mlkl* (duodenum, Appendix A), *Ddit4* (jejunum, Appendix A), and *Csp1* (ileum, Appendix A).

While the PC clusters in all regions express *Olfm4* (Figure 6D–F), there is another cluster in each region that is clearly *Olfm4*-positive, indicating immature progenitor cells. Unfortunately, we have no data for the *Lgr5* probe/gene, so we cannot confirm stem cells with 100% certainty. These cells show a similar expression profile in all clusters based on the Resolve probes, but the *Olfm4* expression is much lower in the ileal slices. The number of actual stem cells found, which are located between the PCs, is very low, with almost none detected. Most *Olfm4*-positive cells are found to be “transit-amplifying cells”, which can be very clearly seen just above the PCs in the duodenum (Figure 6G), jejunum (Figure 6H), and ileum (Figure 6I).

A clear TC cluster, based on *Dclk1* (and *St18*), can be found only in duodenum slices (everywhere, including the crypts), but not in the jejunum or ileum (Figure 5D–F). This observation matches the detection of these cells in the single-cell dataset and confirms the cell-free analysis (Figure 4D) where the marker was found in the crypts co-localized with PC and stemness markers.

In all regions, there are multiple populations of GCs, based on the expression of *Fcgbp* and other markers. In the duodenum, there are two populations (Figure 6D,G), GC-1 (cluster 18) and GC-2 (cluster 1); in the jejunum, we observe three GC clusters: GC-1 (cluster 20), GC-2 (cluster 2), and GC-3 (cluster 13) (Figure 6E,H); the ileum also has three groups of GCs: GC-1 (cluster 21), GC-2 (cluster 1), and GC-3 (cluster 18) (Figure 6F,I). While we successfully identified GCs in the Resolve samples, all of them were localized in the villi of the intestine. However, upon closer examination of the crypts, we observed some crypt cells, primarily in the PC clusters, expressing the *Fcgbp* GC marker, but they were not annotated as GCs.

## 4. Discussion

Previous studies have suggested that PCs constitute a heterogenous population with diverse functions, related to the specific conditions of the small intestinal compartment (duodenum, jejunum, ileum) in which they reside [19,20,57]. scRNA-seq studies on mouse and human small intestines or organoids provided valuable insights into cellular diversity [19,58,59,60]. While PCs can be readily identified as a cluster in these studies, the identification of subclusters within PCs poses a challenge due to their low abundance, and enrichment tools are needed [19,59]. Nevertheless, earlier studies on PCs revealed differences in antimicrobial function and *Defa21* gene expression, depending on the location in the small intestine [19,20]. Recently, two studies were published that managed to identify two subtypes of PCs [19,59]. Grün and colleagues conducted scRNA-seq on mouse organoids and utilized RaceID, an algorithm focusing on identifying rare cell types within complex populations. Using this tool, they identified two PC subtypes: early and late PCs, whereby the early subtype exhibited higher proportions of Lgr5-positive cells than the late ones [59]. Haber et al. [19] were pioneers in performing scRNA-seq on mouse small intestine cells, revealing signature genes for all IECs. Notably, they identified 82 signature genes for PCs using an enrichment strategy for these cells, increasing their proportion to 13.9%. They concluded there are two subtypes of PCs, namely PCs enriched in the proximal small intestine and cells enriched in the distal part, with the former showing high expression of *Rnase1* and the latter having higher expression of α-defensins.

In our study, we present a comprehensive overview of the PC subpopulations in vivo in mice using a triple approach: (i) using a PC purification FACS strategy, we isolated PCs from the entire small intestine for scRNA-seq; (ii) PCs were isolated from the duodenum, jejunum, and ileum for bulk RNA-seq, enabling the identification and investigation of specific PC subpopulations; and (iii) finally, we complemented our data with spatial transcriptomics on tissue section slices of the three different regions, using probes enriched for PC marker genes, allowing us to correlate PC subpopulations with specific intestinal regions.

We confirmed that the FACS gating strategy resulted in a pure PC population (of over 15,000 cells) without significant contamination of non-PCs, as 99.94% expressed PC-specific transcriptomic marker *Lyz1* and at least two members from the α-defensin gene family. In this way, we confirmed the absence of contaminants in the scRNA-seq and investigated the subpopulations present in the overall PC marker-positive population. To acquire and sequence these 15,000 cells, we had to combine material from three mice. This approach provided a broad view on PCs and increased the chance of recovering the less abundant subsets of these cells.

This study was performed using female inbred and young C57BL/6J mice, which was a strategic choice, but it could have caused a slight form of bias, in the sense that mice from other inbred lines might have produced somewhat different results.

We identified several distinct clusters, including the early and late PCs (as described by Grün et al. [59]). In our case, however, the early PCs exhibited only moderately elevated *Lgr5* levels, but they showed a high *Olfm4* signal, which is also associated with stemness. Interestingly, these cells had high expression of ribosomal protein mRNAs, suggesting an expansion of their protein synthesis machinery. This might imply they were in a preparatory phase for their eventual role as secretory cells.

In addition, we found two clusters that expressed clear but lower PC marker expression compared to all the other groups, as well as high expression of markers specific to a different lineage: (i) *Dclk1*-positive cells indicating a tuft cell-like PC identity and (ii) *Fcgbp*-positive cells indicating a GC-like PC phenotype. It is possible that these *Dlck1* and *Fcgbp*-expressing cells are in an intermediate transitional state from stem cell to terminally differentiated goblet and tuft cell, respectively. PCs, GCs, and intestinal tuft cells may represent divergent branches of the same differentiation lineage [61], in which stem cells give rise to a precursor that can then follow the tuft, goblet, or PC differentiation path [23,62]. This means that the current cluster may represent (1) a precursor of the secretory cell lineage which later diverges into distinct branches of GCs/TCs or (2) a differentiated cell type that expresses both GCs and PCs markers, as previously described [63]. However, as our current RNA-seq dataset (bulk as well as single-cell) lacks stem cells or differentiated goblet or tuft cells, we can only hypothesize and refer to follow-up studies.

To understand the basis of the multiple clusters (identified via single-cell analysis) in relation to the origin of the PCs (duodenum, jejunum, ileum), we opted to perform bulk RNA-seq, rather than scRNA-seq, on PCs derived from these locations. We expected this choice to provide a more in-depth understanding of these populations since scRNA-seq is considerably more shallow than bulk RNA-seq. Using this detailed transcriptomic dataset, we confirmed what was described and suggested in the literature [19,20] about the proximal–distal axis and the relation to PC gene expression and function. We were able to provide several additional insights, such as the reduction in the *Olfm4*-positive population along the small intestine and the digestive support functions clearly associated with duodenal and jejunal PCs. By combining the scRNA-seq and bulk RNA-seq information, we annotated most of the clusters observed in our data, with those expressing mitochondrial genes as discriminating markers being the exception. Moreover, we validated the existence of five scRNA-seq clusters associated with the different locations in the small intestine.

To validate these data and potentially establish connections between PC clusters and differences within the crypts, we performed spatial transcriptomics using the Resolve molecular cartography platform. This technology enabled the sequential multiplex measurement of up to 100 genes by means of FISH. The panel design for the spatial transcriptomics was limited by the technology used. We encountered limitations in incorporating the most effective PC markers because of expression levels that were too high according to the platform (*Lyz1*), which could also impede the measurement of other probes. Additionally, in the case of α-defensins, the markers were too short to design suitable probes. This, together with a maximal allowed total (predicted) signal intensity of the probes, limited us to the use of only 59 probes. We also used an in-house pipeline to process the images, segment the cells, and assign the transcripts, which is not possible with Resolve itself. This allowed us to identify PCs and dividing cells, but not to recover a pure stem cell group. This may be due to the small size of stem cell nuclei, which makes them hard to segment. This is also suggested by the occurrence of gaps in the crypts after cell segmentation. We were able to show the tuft cell markers and GC markers co-localizing with PC markers in the crypts, although this is a quite rare event. Furthermore, we were able to validate the classification of early and late PCs along the proximal–distal axis, with decreasing expression of *Olfm4* in PCs along the small intestine. However, we could not detect more than one PC population in a tissue region or within a crypt with the probe set used.

We studied human data to ascertain if the regional α-defensin expression profiles in mice can be translated to humans. However, α-defensin biology is difficult to compare due to the low number of intestinal α-defensin genes in humans. Furthermore, recent studies did not find conclusive differences between PCs in human data [60,64], but this may also be due to the low number of PCs sequenced in human datasets. For example, Burclaff et al. only found 49 PCs to work with. In contrast, there is evidence for the reduction in PC defensin and LYZ1 expression in certain disease conditions such as IBD [65] and liver cirrhosis [66]. For now, this finding in mice cannot be directly translated to humans.

In summary, our study involved a comprehensive in-depth mRNA expression profiling (scRNA-seq, bulk RNA-seq, and spatial transcriptomics) on intestinal PCs across three distinct locations in the small intestine. The analysis unveiled the existence of distinct PC populations along the proximal–distal small intestinal axis, suggesting that the local diverse gut microenvironment and microbiome may determine the transcriptional status of PCs, leading to several distinct PC populations. Through pathway analysis, we identified novel functions of PCs that have not previously been described in the literature. The dataset provides new insights into PCs and introduces numerous potential avenues for further research.

## Figures and Tables

**Figure 1 cells-13-01435-f001:**
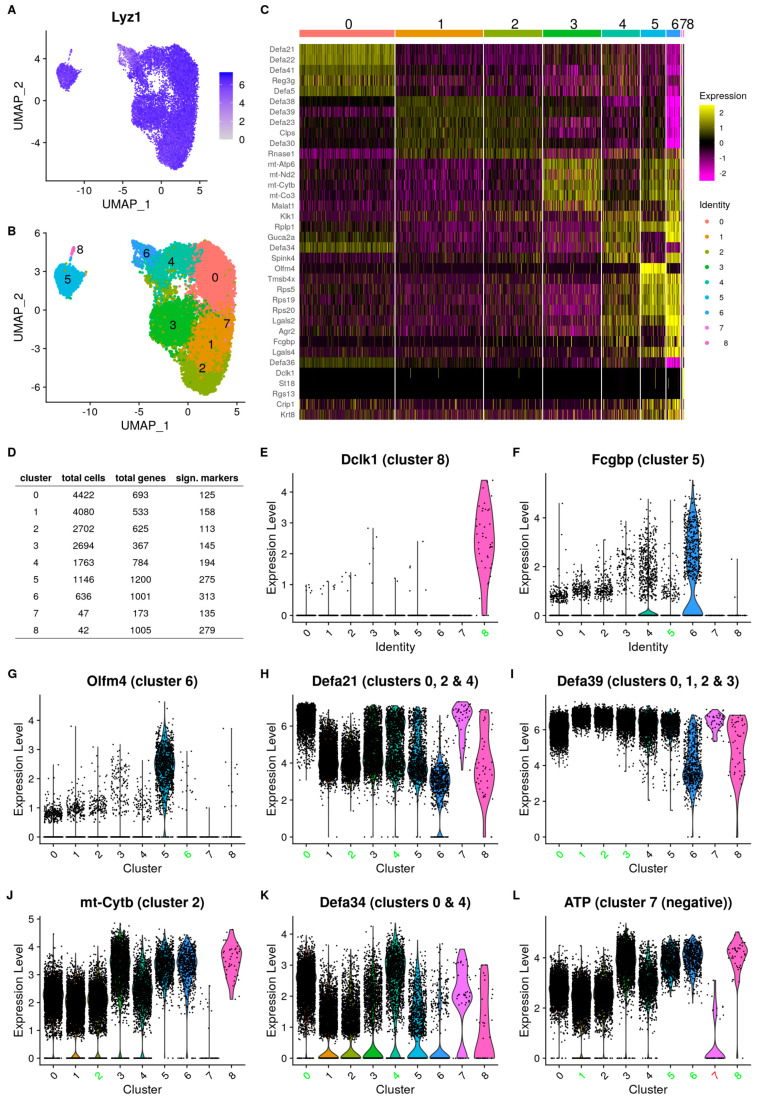
Single-cell transcriptomics analysis on purified PCs. (**A**): Expression plot of *Lyz1* showing that all cells express this typical PC marker gene. (**B**): Uniform manifold approximation and projection (UMAP) of the scRNA-seq dataset, colored according to *Seurat clusters* that are assigned by *Leiden clustering*. In total, 9 clusters were identified. (**C**): Heatmap of the top 5 positive markers per clusters (some markers are shared between clusters). (**D**): Table showing the average numbers of genes detected per cluster and the number of those identified as cluster markers. Some clusters are defined by the number of detected genes in addition to cluster-specific markers (5, 7, and 8). (**E**–**L**): Violin plots of selected marker genes (*Dclk1*, *Fcgbp*, *Olfm4*, *Defa21, Defa39*, *mt-Cytb*, *Defa34*, and *Atp*) depicting the log2-scaled expression levels and the number of cells expressing each marker within each cluster. Red or green cluster numbers indicate in which cluster the markers are downregulated or upregulated, respectively.

**Figure 2 cells-13-01435-f002:**
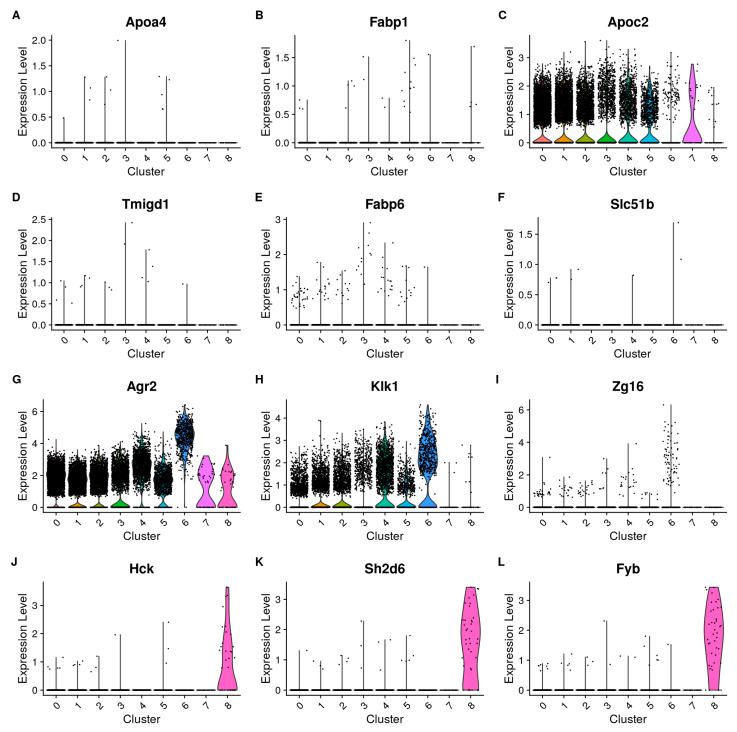
Markers identified by Haber et al. (**A**–**F**): Most enterocyte marker genes (*Apoa4*, *Fabp1*, *Tmigd1*, *Fabp6*, and *Slc51b*) are expressed very little and in only very few cells, with some exceptions (*Apoc2*, **C**). (**G**,**H**): Goblet cell markers (*Agr2* and *Klkl1*) can be found in our dataset and are most highly expressed in cluster 5. (**I**): Goblet cell marker *Zg16* is only found in a few cells, with the majority and the highest expression levels observed in cluster 5. (**J**–**L**): Tuft cell markers (*Hck*, *Sh2d6*, and *Fyb*) described by Haber et al. are indeed expressed in the cells from cluster 8 of our dataset. All clusters express PC markers as well.

**Figure 3 cells-13-01435-f003:**
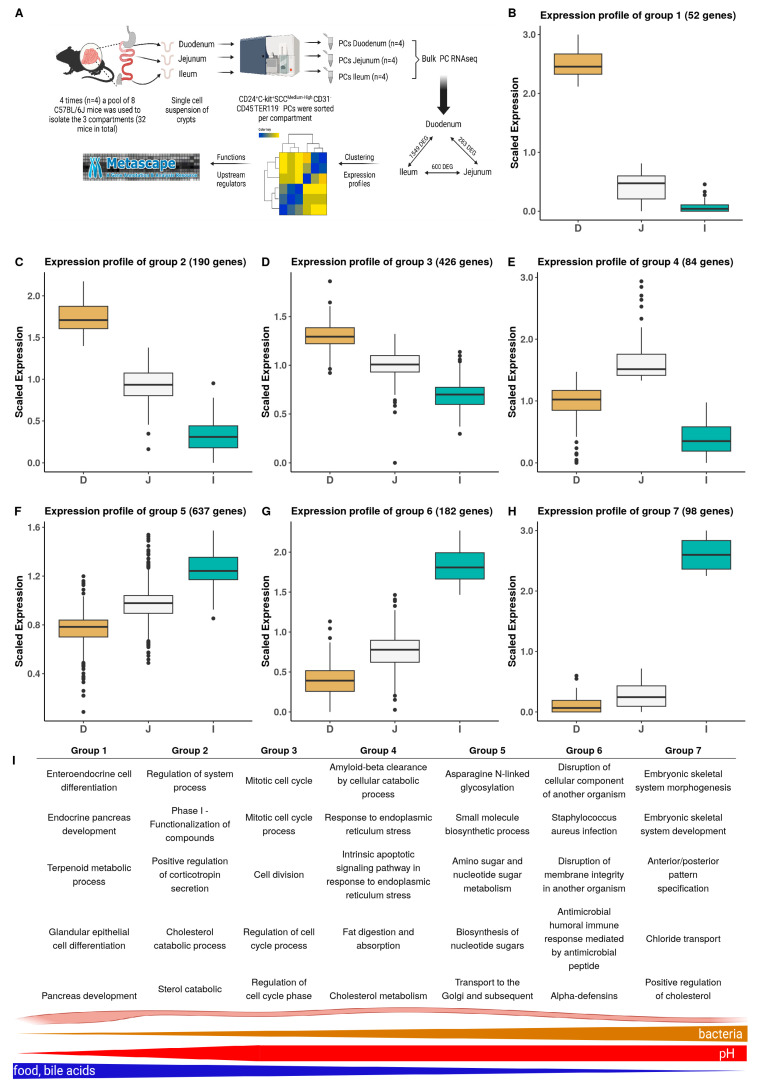
PC bulk RNA-seq analysis across regions of the small intestine. (**A**): Workflow of the bulk RNA-seq experiment. (**B**–**H**): Expression profiles of groups 1–7. (**I**): Top 5 differentially expressed pathways for each group as found via *Metascape* analysis (the expression profiles of the different groups can be roughly placed along the small intestine). The graph under the table shows a schematic representation of the most important proximal-to-distal gradients in the small intestine. Whiskers indicate variability outside the upper and lower quartiles. Outliers, which are data points that fall outside this range, are plotted as individual dots aligned with the whiskers.

**Figure 4 cells-13-01435-f004:**
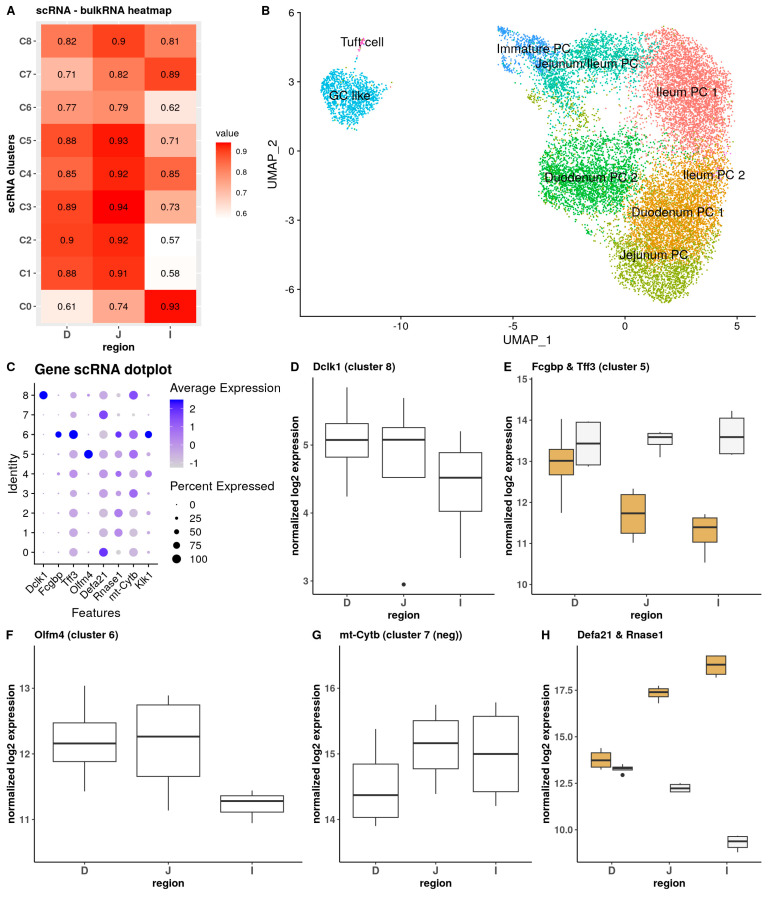
Integration of the scRNA-seq and bulk RNA-seq data. (**A**): Correlation matrix between scRNA-seq clusters and bulk RNA-seq (based on a *pseudo-bulk* expression dataset). (**B**): scRNA-seq UMAP plot with all subpopulations annotated based on the scRNA-seq marker and region bulk RNA-seq data. (**C**): Dotplot showing the scRNA-seq expression of the selected marker genes, described in the following panels. (**D**–**H**): Expression profile of single-cell marker genes in the bulk RNA-seq, *Dclk1* (**D**), *Fcgbp* and *Tff3* (**E**), *Olfm4* (**F**), *Mt-cytb* (**G**), and *Defa21* and *Rnase1* (**H**).

**Figure 5 cells-13-01435-f005:**
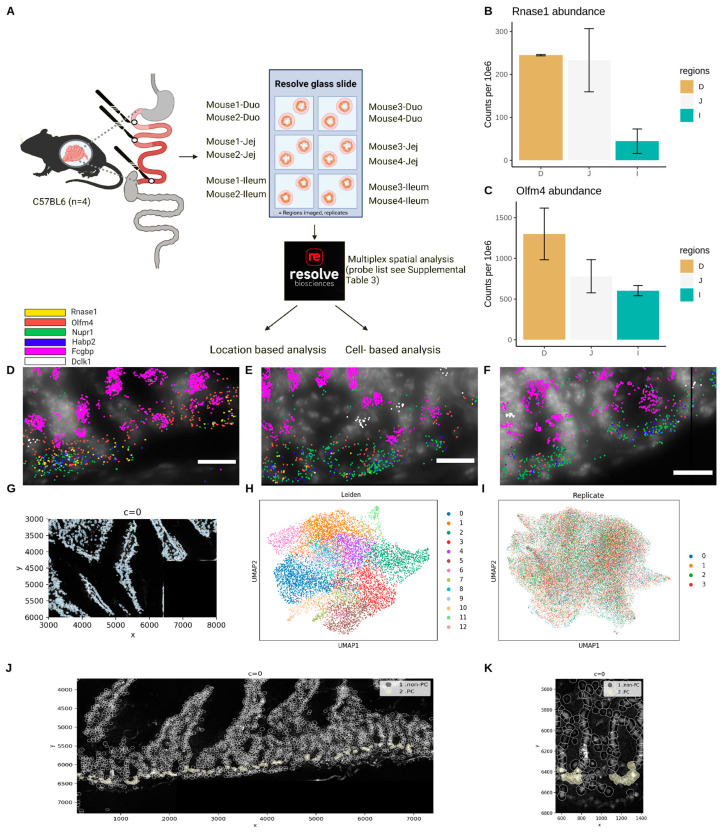
Resolve molecular cartography. (**A**): Workflow of the Resolve spatial transcriptomics experiment. (**B**,**C**): Average abundance of *Rnase1* and *Olfm4* in the Resolve samples (in counts per million) showing a decreasing expression gradient along the small intestine. (**D**–**F**): Markers found in the crypts of the duodenum (**D**), jejunum (**E**), and ileum (**F**). Yellow: *Rnase1*, red: *Olfm4*, blue and green: PC markers (*Hapb2, Nupr1*), purple: *Fcgbp*, white: *Dclk1*. A color code legend is added above panel (**D**). Scalebar 20 µm. PC marker signals are strongest in ileum, whereas *Olfm4* and *Rnase1* exhibit weaker signals. The GC marker *Fcgbp* can also, rarely, be found in crypts where the signal co-localizes with *Olfm4*/*Hapb2*/*Nupr1*. *Dclk1*-positive cells are detected in the crypts in close proximity with *Olfm4*/*Hapb2*/*Nupr1*. (**G**): *Cellpose* nuclei segmentation on duodenum region 1. (**H**): *Leiden clustering* of duodenum region 1. (**I**): Overlap of the biological replicates of the duodenum to search for batch effects. The perfect overlap indicates the absence of batch effects. (**J**): Localizing the PCs (2—yellow) and non-Paneth cells (1—black) within a specific region of the duodenum. The provided image is a representative subset of this region. Full images, as well as images of all other samples, can be found in the Appendix A repository Figshare (10.6084/m9.figshare.25999150). (**K**): A zoomed-in view of the figure shown in panel J, focusing on the region with x-coordinates [540;1410], containing 2 crypts and clearly showing the presence of PCs within these crypts. D: duodenum, J: jejunum, and I: Ileum.

**Figure 6 cells-13-01435-f006:**
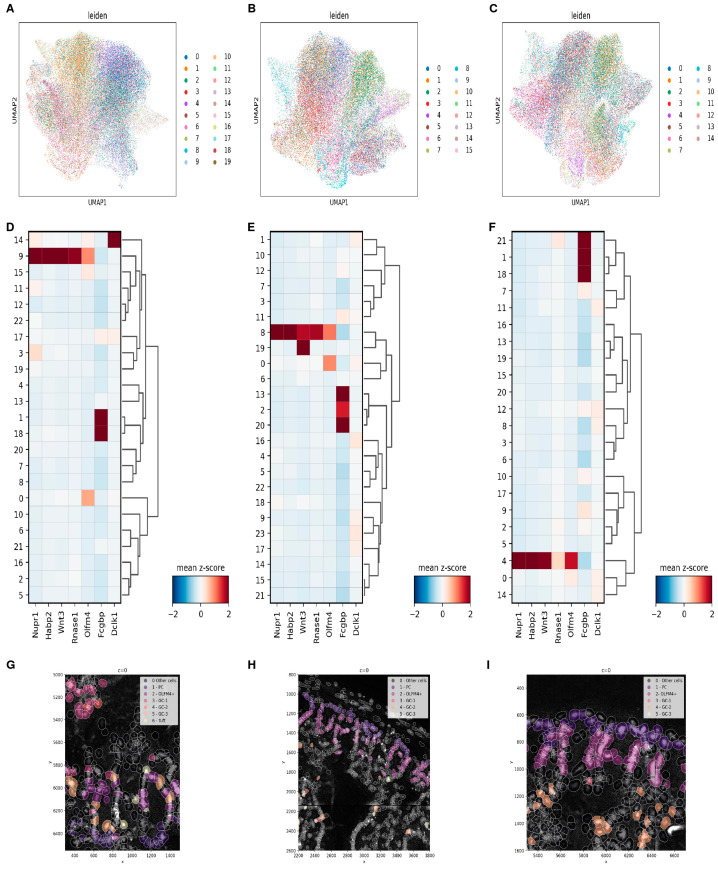
Resolve cell-based analysis. (**A**–**C**): *Leiden clustering* of the merged duodenum samples (**A**), merged jejunum samples (**B**), and merged ileum samples (**C**). (**D**–**F**): Marker expression z-scores for markers of PCs (*Nupr1, Habp2, Rnase1*, and *Wnt3*), immature cells (*Olfm4*), GCs (*Fcgbp*), and tuft cells (*Dclk1*) in duodenum (**D**), jejunum (**E**), and ileum (**F**). (**G**–**I**): PCs, *Olfm4*-positive cells, tuft cells, and GCs in the duodenum (**G**), jejunum (**H**), and ileum (**I**).

## Data Availability

The bulk RNA-seq data were deposited in the NCBI’s Gene Expression Omnibus database (GEO) with number: GSE255507. Resolve images were deposited in the Appendix A repository Figshare (10.6084/m9.figshare.25999150). All RNA-seq data can be found in the public domain with accession numbers GSE255507 (PC bulk RNA-seq) and GSE273983 (PC scRNA-seq).

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
