# Peer review of "Identification and Characterization of Multiple Paneth Cell Types in the Mouse Small Intestine"

_cells, 2024, doi:10.3390/cells13171435_

Round 1

Reviewer 1 Report

Comments and Suggestions for Authors

This paper uses a combination of single cell RNA seq, Bulk RNA seq and spatial transcriptomics to identify different subtypes of Paneth cells in different parts of the intestine. However there are several points that needs to be addressed before considering for publication.

Major points:

1.      For Figure 1E and 1F, the authors indicate that some PCs displays very high gene signature of Tuft cells and Goblet cells. Do high expression of these genes indicate the functional similarity to Tuft or Goblet cells compared to other PCs?

2.      For Fig.1 G, the authors indicate that Cluster 6 PCs showed high expression of stem cell markers such as OLFM4 and Lgr5. Is it possible that these cells are actually transient amplifying cells but not real PCs?

3.      In Fig.3B-3H, what do the dots mean in the bar graph? Some of the bars didn’t have them but some do. Please explain more details in the figure of legends. Besides there’s no statistics.

4.      For Supplementary Fig.4, all the words are so small, and the pictures are very low resolution that are really hard to read and understand. Please increase the font sizes and replace with higher quality images.

5.      For Fig.4C, the authors indicate that there are several Defa genes showing the pattern as Defa 21. It might be better to add the expression of more Deta genes in duodenum, jejunum and ileum to make the point stronger.

6.      In Fig 5D-F, it might be better to add canonical methods such as IF staining of OLFM4, RNase1 and HaPb2 in Duodenum, Jejunum and Ileum to further validate the result if this is doable.

7.      Fig 6G-I: it is very hard to see the transient amplifying cells through the image. Please use a higher resolution image and higher magnification.

Minor points:

1.      For Figure S2, please replace these charts with higher resolution figures. It’s very hard to read and after magnification it is very blurry.

2.      Page 10 paragraph 2: It might be better to add references for the learnt knowledge about Defa genes and their location.

3.      For Fig. 3B-H, please enlarge the D, J, I at the bottom of the X axis since they are very hard to read.

4.      Please consider moving Fig 4C to the end of Fig.4 since the logic is a little messy if it is mentioned in the last paragraph of the session.

5.      For Fig. 5D-F, please add the label of the corresponding marker with different colors. For the whole Fig.5 please use higher resolution images and larger fonts for labeling since it is not readable.

Author Response

Reviewer one

This paper uses a combination of single cell RNA seq, Bulk RNA seq and spatial transcriptomics to identify different subtypes of Paneth cells in different parts of the intestine. However there are several points that needs to be addressed before considering for publication.

Major points:

Comments 1: For Figure 1E and 1F, the authors indicate that some PCs displays very high gene signature of Tuft cells and Goblet cells. Do high expression of these genes indicate the functional similarity to Tuft or Goblet cells compared to other PCs?

Response 1:

In our opinion, these tuft cell-like PCs or GC-like PCs can represent (1) a differentiated Paneth cell type with Tuft cell or Goblet cell characteristics or (2) a precursor or intermediate transitional state of the secretory cell lineage that still have the same differentiation path, which later diverge into distinct branches.

These hypotheses suggest that the high expression of Tuft cell and Goblet cell genes in some PCs could reflect either a unique, multipotent cell type or a transitional state in the differentiation process. However, the challenge to investigate this question is threefold: (1) the limited starting material, (2) the lack of an existing sorting protocol for differentiating between Paneth cell populations and (3) that our current RNA-seq datasets (bulk as well as single cell) lack differentiated goblet or tuft cells. Further studies are needed to elucidate the exact nature of these cells.

This was already briefly addressed in the discussion, but we elaborated more on this to address the reviewer's question comprehensively.

Discussion - Line 750-762: In addition, we found two clusters that expressed clear but lower PC marker expression compared to all the other groups, as well as high expression of markers specific to a different lineage: (i) Dclk1 positive cells indicating a tuft cell-like PC identity and (ii) Fcgbp positive cells indicating a GC-like PC phenotype. It is possible that these Dlck1 and Fcgbp expressing cells are in an intermediate transitional state from stem cell to terminally differentiated goblet and tuft cell, respectively. PCs, GCs and intestinal tuft cells may represent divergent branches of the same differentiation lineage1, in which stem cells give rise to a precursor that can then follow the tuft, goblet or PC differentiation path2,3. This means that the current cluster may represents (1) a precursor of the secretory cell lineage which later diverge into distinct branches of GCs/TCs or (2) a differentiated cell type that both express GCs and PCs markers, as previously described4. However, as our current RNA-seq dataset (bulk as well as single cell) lack stem cells or differentiated goblet or tuft cells, we can only hypothesize and refer to follow-up studies.

Comments 2: For Fig.1 G, the authors indicate that Cluster 6 PCs showed high expression of stem cell markers such as OLFM4 and Lgr5. Is it possible that these cells are actually transient amplifying cells but not real PCs?

Response 2: Thank you for pointing this out. Intestinal stem cells in the lower part of the crypt indeed give rise to a larger pool of transit-amplifying cells that can then differentiate into secretory or absorptive progenitors. Our identified immature PC population already exhibits secretory characteristics and expresses clear PC markers. Therefore, it is more accurate to refer to these cells as immature PC progenitor cells, rather than immature PCs. We have made this adjustment in the manuscript.

Comments 3:  In Fig.3B-3H, what do the dots mean in the bar graph? Some of the bars didn’t have them but some do. Please explain more details in the figure of legends. Besides there’s no statistics.

Response 3: Thank you for your question. In Figure 3B-3H, the lines extending parallel from the boxes are known as the “whiskers.” These whiskers indicate variability outside the upper and lower quartiles. Outliers, which are data points that fall outside this range, are plotted as individual dots aligned with the whiskers.

We will update the figure legends to include this explanation. We did not include statistics as this is the representation of the expression profile of the groups from the k-means clustering algorithm. It is not required that the D, J & I are different in any cluster, only that the clusters differ from each other.

Comments 4: For Supplementary Fig.4, all the words are so small, and the pictures are very low resolution that are really hard to read and understand. Please increase the font sizes and replace with higher quality images.

Response 4: We have remade Figure S4 in a different layout and panel format to improve readability.

Comments 5: For Fig.4C, the authors indicate that there are several Defa genes showing the pattern as Defa 21. It might be better to add the expression of more Deta genes in duodenum, jejunum and ileum to make the point stronger.

Response 5: We agree with the reviewer. We have created a new figure (Figure S6) that shows the expression of the 16 overall highest expressed Defa genes in the bulk RNA-seq data across the different compartments. This analysis highlights several genes that follow the Defa21 expression pattern, along with defensins that follow different expression patterns. We have described this in more detail in the text.

Line 539-545: We also examined overall α-defensin expression in more detail. Among the 16 overall highest expressed α-defensin genes, we identified five genes that share the same expression pattern as Defa21 (including Defa21 itself) and two additional genes, Defa29 and Defa5 that exhibit a closely related pattern (with lower expression in the duodenum). Several genes, such as Defa40 and Defa3, show a peak in expression in the jejunum. Additionally, we observed that some genes, e.g. Defa39 and Defa38 have expression patterns nearly opposite to that of Defa21 (Fig S6).

Comments 6: In Fig 5D-F, it might be better to add canonical methods such as IF staining of OLFM4, RNase1 and HaPb2 in Duodenum, Jejunum and Ileum to further validate the result if this is doable.

Response 6: We agree that IF staining would be valuable for validation on the protein level. However, performing multiplex IHC staining is quite challenging. Specifically, we need to find these three different antibodies raised in three different species, and each antibody requires an individually optimized protocol. Currently, antibodies for HaPb2 and RNase1 against mice are only available raised in rabbit as host species, making multiplexing difficult since secondary antibodies must be species-specific and distinguishable. Due to these practical limitations for IHC multiplexing, we opted for a multiplex method that allows us to measure transcripts of up to 59 probes simultaneously.

Comments 7:  Fig 6G-I: it is very hard to see the transient amplifying cells through the image. Please use a higher resolution image and higher magnification. 

Response 7: Thanks for the remark, we adapted the figure as suggested. We have provided a more detailed, zoomed-in view of the intestinal zones, highlighting the crypts and villi. While this focus enhances detail, it removes the broader overview. For that reason, we have included a direct DOI to the full-size images: 10.6084/m9.figshare.25999150.

Line 872-875 in the Data Availability Statement: Resolve images are deposited in the supplemental data repository Figshare (10.6084/m9.figshare.25999150).

Minor points:

Comments 8: For Figure S2, please replace these charts with higher resolution figures. It’s very hard to read and after magnification it is very blurry.

Response 8: We agree with this comment. We remade the figure to the same panel composition and overall layout like Figure S4 for improved readability and consistency.

Comments 9: Page 10 paragraph 2: It might be better to add references for the learnt knowledge about Defa genes and their location.

Response 9: Thanks for the suggestion. We added the reference about the defa knowledge and their location.

Line 319-323: Among the IECs, PCs are unique in the expression of “Defa gene family members”, a set of highly conserved 2-exon genes that encode α-defensins with antimicrobial functions5,6. The genes, numbered from Defa1 to Defa43, are located on chromosome 87. However, the numbering is not continuous, and 39 Defa genes are annotated in the MGI database.

Comments 10: For Fig. 3B-H, please enlarge the D, J, I at the bottom of the X axis since they are very hard to read.

Response 10: Indeed, we made the labels more clear.

Comments 11: Please consider moving Fig 4C to the end of Fig.4 since the logic is a little messy if it is mentioned in the last paragraph of the session.

Response 11: Thanks for the suggestion. We have moved panel C to the end of Figure 4 and reordered the panels. This adjustment aligns the structure of the figure with the structure of the text.

Comments 12: For Fig. 5D-F, please add the label of the corresponding marker with different colors. For the whole Fig.5 please use higher resolution images and larger fonts for labeling since it is not readable.

Response 12: We enhanced the figures by adding colored labels and incorporating the suggested adaptations.

SOURCES

  1. Yang, Q., Bermingham, N.A., Finegold, M.J., and Zoghbi, H.Y. (2001). Requirement of Math1 for secretory cell lineage commitment in the mouse intestine. Science (80-. ). 294, 2155–2158. https://doi.org/10.1126/science.1065718.
  2. Wallaeys, C., Garcia‐Gonzalez, N., and Libert, C. (2023). Paneth cells as the cornerstones of intestinal and organismal health: a primer. EMBO Mol. Med. 15, 1–26. https://doi.org/10.15252/emmm.202216427.
  3. Kolev, H.M., and Kaestner, K.H. (2023). Mammalian Intestinal Development and Differentiation—The State of the Art. Cmgh 16, 809–821. https://doi.org/10.1016/j.jcmgh.2023.07.011.
  4. Nyström, E.E.L., Martinez-Abad, B., Arike, L., Birchenough, G.M.H., Nonnecke, E.B., Castillo, P.A., Svensson, F., Bevins, C.L., Hansson, G.C., and Johansson, M.E.V. (2021). An intercrypt subpopulation of goblet cells is essential for colonic mucus barrier function. Science (80-. ). 372, eabb1590. https://doi.org/10.1126/science.abb1590.
  5. Huttner, K.M., Selsted, M.E., and Ouellette, A.J. (1993). Structure and diversity of the murine cryptdin gene family. Genomics 19, 448–453.
  6. Jones, D.E., and Bevins, C.L. (1992). Paneth cells of the human small intestine express an antimicrobial peptide gene. J. Biol. Chem. 267, 23216–23225. https://doi.org/10.1016/s0021-9258(18)50079-x.
  7. Amid, C., Rehaume, L.M., Brown, K.L., Gilbert, J.G.R., Dougan, G., Hancock, R.E.W., and Harrow, J.L. (2009). Manual annotation and analysis of the defensin gene cluster in the C57BL/6J mouse reference genome. BMC Genomics 10, 1–13. https://doi.org/10.1186/1471-2164-10-606

Reviewer 2 Report

Comments and Suggestions for Authors

In this manuscript, Steven Timmermans and colleagues provide a detailed investigation into the Paneth cells (PCs) and heterogeneous populations in mice using multiple approaches with varying depths. They identified several distinct clusters of mature and immature PCs, which expressed universal PC markers as well as specific genes of goblet cells, tuft cells, intestinal stem cells (ISCs), etc., implying they may have different functions in the intestine. Additionally, PCs from different segments showed local diversity in the gut microenvironment and microbiome, influencing the transcriptional status of PCs. Overall, this study is interesting and comprehensive; however, some details should be addressed:

  1. Do the authors observe similar expression patterns of markers from different types of PCs in various segments of the human intestine? This can be addressed using published datasets.
  2. KEGG or GO analysis of the differential genes from different segments of PCs in mice can supplement and verify the data from Figure 3.
  3. The subtitles in the results section should be reorganized. The authors should state the main findings from the data instead of describing the analysis or experiments they performed.
  4. Scale bars should be provided in Figures 5D-F, G, J-K, and 6G-I.
  5. High-resolution images should be provided for Figures 6G-I.
  6. The data visualization is clear and straightforward, but the sc-RNAseq data availability should be included in the Data Availability Statement.
  7. Given that the authors are analyzing PCs, details of the sc-RNAseq processing should be provided, such as the number of PCs used for clustering the cells and UMAP, the resolution, and key package versions.
  8. The regional differences in the gut are quite interesting. There are no references supporting the statement in lines 45-48. Two follow-up studies focused on the regional differences among intestine segments and identified key location-dependent transcripts (PMIDs: 38678016 and 34582804) that could be cited here.
  9. The most known function of PCs is their niche function for ISCs and their antimicrobial ability. Another function of PCs is their preferential generation from ISCs to exert their niche function during intestinal repair following irradiation. The authors could supplement this information in lines 71-72.
  10. Line 371 can be part of the Data Availability Statement.

Author Response

Reviewer two:

In this manuscript, Steven Timmermans and colleagues provide a detailed investigation into the Paneth cells (PCs) and heterogeneous populations in mice using multiple approaches with varying depths. They identified several distinct clusters of mature and immature PCs, which expressed universal PC markers as well as specific genes of goblet cells, tuft cells, intestinal stem cells (ISCs), etc., implying they may have different functions in the intestine. Additionally, PCs from different segments showed local diversity in the gut microenvironment and microbiome, influencing the transcriptional status of PCs. Overall, this study is interesting and comprehensive; however, some details should be addressed:

Comments 1: Do the authors observe similar expression patterns of markers from different types of PCs in various segments of the human intestine? This can be addressed using published datasets.

Response 1: This is an interesting remark. We have incorporated in the discussion the recent studies on human material at the single cell level. Based on the results reported in literature so far, there is no evidence for regional differences in α-defensin expression in the small intestine. However, due to the differences in α-defensin gene numbers and the low amount of cells investigated in human dataset, we cannot exclude that the differences found in mice are not there. Differences in such can be found between baseline and diseases (e.g. IBD, cirrhosis).

Line 795-802: We studied human data to ascertain if the regional α-defensin expression profiles in mice can be translated to human. However, α-defensin biology is difficult to compare due to the low amount of intestinal α-defensin genes in humans. Furthermore recent studies did not found conclusive differences between PCs in human data1,2, but this may also be due to the low number of PCs sequenced in human datasets. E.g. Burclaff et al. only found 49 PCs to work with. In contrast, there is evidence for reduction of PC defensin and LYZ1 expression in certain disease conditions such as IBD3 and liver cirrhosis4. So for now, this finding in mice cannot be directly translated to humans.

Comments 2: KEGG or GO analysis of the differential genes from different segments of PCs in mice can supplement and verify the data from Figure 3.

We agree. KEGG and GO analysis were already provided in Figure S4. We also added an extra supplemental table as a multipage excel file, so that the reader can see which genes are linked with the identified pathways. 

Comments 3: The subtitles in the results section should be reorganized. The authors should state the main findings from the data instead of describing the analysis or experiments they performed.

Thanks you for pointing this out. We adapted all the titles in the result section to describe the main findings, rather than the used technique.

Comments 4: Scale bars should be provided in Figures 5D-F, G, J-K, and 6G-I.

High-resolution images should be provided for Figures 6G-I.

We added the scalebars  to most of the figures and provided high-resolution images. Only for the images created with the internal pipeline, we were not able to provide scalebars, as the software does not support this.

Comments 5: The data visualization is clear and straightforward, but the sc-RNAseq data availability should be included in the Data Availability Statement.

We agree with this comment. We added following sentence to the Data Availability Statement:

All RNA-seq data can be found in the public domain with accession number GSE255507 (PC bulk RNA-seq) and GSE273983 (PC scRNA-seq).

Comments 6: Given that the authors are analyzing PCs, details of the sc-RNAseq processing should be provided, such as the number of PCs used for clustering the cells and UMAP, the resolution, and key package versions.

Thank you for the comment. This information is now provided in the material and methods sections and in the result section. 

Comments 7: The regional differences in the gut are quite interesting. There are no references supporting the statement in lines 45-48. Two follow-up studies focused on the regional differences among intestine segments and identified key location-dependent transcripts (PMIDs: 38678016 and 34582804) that could be cited here.

Thanks for the good suggestion, we added all the sources about regional differences and cited both suggested paper as well.

Along these segments, gradients of distinct environmental conditions have been observed, for example the amounts of bacteria5 and their diversity and composition6, digestive enzymes7, metabolites8 etc. Recent research studying regional variations along the gastrointestinal tract has identified a crucial role of location-specific transcripts9,10.

Comments 8: The most known function of PCs is their niche function for ISCs and their antimicrobial ability. Another function of PCs is their preferential generation from ISCs to exert their niche function during intestinal repair following irradiation. The authors could supplement this information in lines 71-72.

We agree, the paper from Schmitt et al. should be mentioned here. We added this information to the paper. line 73-75:

Line Paneth cells can contribute to the intestinal regenerative response to irradiation by de-differentiation and gaining stem cell-like characteristics.

Comments 9: Line 371 can be part of the Data Availability Statement.

Response 9: We removed Line 371, and placed the information in the Data Availability Statement.

SOURCES

  1. Burclaff, J., Bliton, R.J., Breau, K.A., Ok, M.T., Gomez-Martinez, I., Ranek, J.S., Bhatt, A.P., Purvis, J.E., Woosley, J.T., and Magness, S.T. (2022). A Proximal-to-Distal Survey of Healthy Adult Human Small Intestine and Colon Epithelium by Single-Cell Transcriptomics. Cmgh 13, 1554–1589. https://doi.org/10.1016/j.jcmgh.2022.02.007.
  2. Hickey, J.W., Becker, W.R., Nevins, S.A., Horning, A., Perez, A.E., Zhu, C., Zhu, B., Wei, B., Chiu, R., Chen, D.C., et al. (2023). Organization of the human intestine at single-cell resolution. Nature 619, 572–584. https://doi.org/10.1038/s41586-023-05915-x.
  3. Ramasundara, M., Leach, S.T., Lemberg, D.A., and Day, A.S. (2009). Defensins and inflammation: The role of defensins in inflammatory bowel disease. J. Gastroenterol. Hepatol. 24, 202–208. https://doi.org/10.1111/j.1440-1746.2008.05772.x.
  4. Muñoz, M., Eidenschenk, C., Ota, N., Wong, K., Lohmann, U., Kühl, A.A., Wang, X., Manzanillo, P., Li, Y., Rutz, S., et al. (2015). Interleukin-22 Induces Interleukin-18 Expression from Epithelial Cells during Intestinal Infection. Immunity 42, 321–331. https://doi.org/10.1016/j.immuni.2015.01.011.
  5. Sender, R., Fuchs, S., and Milo, R. (2016). Revised Estimates for the Number of Human and Bacteria Cells in the Body. PLoS Biol. 14, 1–14. https://doi.org/10.1371/journal.pbio.1002533.
  6. Yersin, S., and Vonaesch, P. (2024). Small intestinal microbiota: from taxonomic composition to metabolism. Trends Microbiol., 1–14. https://doi.org/10.1016/j.tim.2024.02.013.
  7. Layer, P., Go, V.L.W., and DiMagno, E.P. (1986). Fate of pancreatic enzymes during small intestinal aboral transit in humans. Am. J. Physiol. - Gastrointest. Liver Physiol. 251, 475–480. https://doi.org/10.1152/ajpgi.1986.251.4.g475.
  8. Folz, J., Culver, R.N., Morales, J.M., Grembi, J., Triadafilopoulos, G., Relman, D.A., Huang, K.C., Shalon, D., and Fiehn, O. (2023). Human metabolome variation along the upper intestinal tract. Nat. Metab. 5, 777–788. https://doi.org/10.1038/s42255-023-00777-z.
  9. Gu, W., Wang, H., Huang, X., Kraiczy, J., Singh, P.N.P., Ng, C., Dagdeviren, S., Houghton, S., Pellon-Cardenas, O., Lan, Y., et al. (2022). SATB2 preserves colon stem cell identity and mediates ileum-colon conversion via enhancer remodeling. Cell Stem Cell 29, 101-115.e10. https://doi.org/10.1016/j.stem.2021.09.004.
  10. Gu, W., Huang, X., Singh, P.N.P., Li, S., Lan, Y., Deng, M., Lacko, L.A., Gomez-Salinero, J.M., Rafii, S., Verzi, M.P., et al. (2024). A MTA2-SATB2 chromatin complex restrains colonic plasticity toward small intestine by retaining HNF4A at colonic chromatin. Nat. Commun. 15, 1–16. https://doi.org/10.1038/s41467-024-47738-y.

Round 2

Reviewer 1 Report

Comments and Suggestions for Authors

The authors addressed all the points I raised properly already. 

Reviewer 2 Report

Comments and Suggestions for Authors

The authors have successfully addressed all my concerns, this manuscript is now ready for publication.